# Multilingual Holistic Bias: Extending Descriptors and Patterns to Unveil Demographic Biases in Languages at Scale

**Marta R. Costa-jussà, Pierre Andrews, Eric Smith, Prangthip Hansanti, Christophe Ropers, Elahe Kalbassi, Cynthia Gao, Daniel Licht, Carleigh Wood**

†FAIR, Meta

{costajussa,mortimer,ems,prangthiphansanti,
chrisropers,ekalbassi,cynthiagao,dlicht,carleighwood}@meta.com

## Abstract

We introduce a multilingual extension of the HolisticBias dataset, the largest English template-based taxonomy of textual people references: Multilingual HolisticBias. This extension consists of 20,459 sentences in 50 languages distributed across 13 demographic axes. Source sentences are built from combinations of 118 demographic descriptors and three patterns, excluding nonsensical combinations. Multilingual translations include alternatives for gendered languages that cover gendered translations when there is ambiguity in English. Our dataset is intended to uncover demographic imbalances and be the tool to quantify mitigations towards them.

Our initial findings show that translation quality for EN-to-XX translations is an average of almost 8 spBLEU better when evaluating with the masculine human reference compared to feminine. In the opposite direction, XX-to-EN, we compare the robustness of the model when the source input only differs in gender (masculine or feminine) and masculine translations are an average of almost 4 spBLEU better than feminine. When embedding sentences to a joint multilingual sentence representations space, we find that for most languages masculine translations are significantly closer to the English neutral sentences when embedded.

*WARNING: the current paper contains examples that may be offensive.*

## 1 Introduction

Demographic biases are relatively infrequent phenomena but present a very important problem. The development of datasets in this area has raised the interest in evaluating Natural Language Processing (NLP) models beyond standard quality terms.

This can be illustrated by the fact that machine translation (MT) models systematically translate neutral source sentences into masculine or feminine depending on the stereotypical usage of the word

Figure 1: Pathological examples (in sentence piece) of Multilingual HolisticBias source and NLLB translation, with Spanish masculine/feminine references. Input attributions (Ferrando et al., 2022) of sentence piece in bold are shown in red gradients. (1) English is translated into Spanish, and we observe masculine form, which could be overgeneralisation or a stereotype. (2) English is translated into Spanish masculine for "amigos", which can be seen an overgeneralisation, but into feminine for "amas de casa", which is a stereotype. (3) is translated into feminine, which given the lack of translations into feminine, we assume is a stereotypical translation. (4) Spanish masculine/feminine human translations used as source, NLLB translations and English as reference. These examples illustrate the lack of gender robustness, supported by similar input attributions.

(e.g. "homemakers" into "amas de casa", which is the feminine form in Spanish and "doctors" into "médicos", which is the masculine form in Spanish).

While gender is one aspect of demographic biases, we can further explore abilities, nationalities, races or religion and observe other generalizations of the models that may perpetuate or amplify stereotypes and inequalities in society. Quantifying and evaluating these biases is not straightforward because of the lack of datasets and evaluation metrics. Proper evaluation will enable mitigation of these biases.

**Related work** HolisticBias (Smith et al., 2022) is an English dataset built from templated sentences that can elicit enough examples in various contexts to analyze and draw actionable conclusions: when measuring toxicity after translating HolisticBias prompts (Costa-jussà et al., 2023); when measuring the relative perplexity of different sentences as a function of gendered noun or descriptor (Smith et al., 2022); when looking at skews of the usages of different descriptors in the training data, etc. Other datasets consisting of slotting terms into templates were introduced by (Kurita et al., 2019; May et al., 2019; Sheng et al., 2019; Brown et al., 2020; Webster et al., 2020), to name a few. The advantage of templates is that terms can be swapped in and out to measure different forms of social biases, such as stereotypical associations. Other strategies for creating bias datasets include careful hand-crafting of grammars (Renduchintala and Williams, 2022), collecting prompts from the beginnings of existing text sentences (Dhamala et al., 2021), and swapping demographic terms in existing text, either heuristically (Papakipos and Bitton, 2022) or using trained neural language models (Qian et al., 2022). Most of these alternatives cover few languages or they are limited in the bias scope (e.g. only gender (Stanovsky et al., 2019; Renduchintala et al., 2021; Levy et al., 2021; Costa-jussà et al., 2022; Renduchintala and Williams, 2022; Savoldi et al., 2021; **?**)).

**Contributions** Our work approaches this problem by carefully translating a subset of the HolisticBias dataset into 50 languages (see appendix A for a complete list), covering 13 demographic axes. As an extension of HolisticBias, we will invite additions and amendments to the dataset, in order to contribute to its establishment as a standardized method for evaluating bias for highly multilingual NLP models. We use the proposed dataset to experiment on MT and sentence representation. Results when translating from English show an average of almost 8 spBLEU reduc-

tion when evaluating on the feminine reference set compared to masculine. This showcases the preference towards masculine translations. Among the 13 demographic axes of HolisticBias, the quality of translation averaged across languages is highest for the nationality axis and lowest for the cultural axis. Results when translating to English show that the masculine set has almost 4 spBLEU improvement compared to the feminine set. When embedding sentences to a joint multilingual sentence representations space which is the core tool of multilingual data mining, we find that for most languages, there is a significant difference in the similarity between the masculine translations and the feminine one. Masculine translations are significantly closer to the English sentence when embedded, even if this difference remains small and we do not yet know the effect on the mining algorithm.

## 2 Background: HolisticBias

HolisticBias is composed of 26 templates, more than 600 descriptors (covering 13 demographic axes) and 30 nouns. Overall, this dataset consists of over 472k English sentences used in the context of a two-person conversation. Sentences are typically created from combining a sentence template (e.g., "I am a [NOUN PHRASE]."), a noun (e.g., parent), and a descriptor (e.g., disabled). The list of nearly 600 descriptors covers 13 demographic axes such as ability, race/ethnicity, or gender/sex. The noun can imply a certain gender (e.g. woman, man) or avoid gender references (e.g. child, kid). Sentence templates allow for both singular and plural forms of the descriptor/noun phrase.

Experiments in MT with the NLLB model using the full initial (English only) version of the HolisticBias dataset, as reported in (Costa-jussà et al., 2023), show that the percentage of true added toxicity is also relatively low (from 0.004% in Chinese to 1.07% in Kinyarwanda) but that the number of examples in absolute value is much greater (20 in Chinese, 4,951 in Kinyarwanda) due to the fact that HolisticBias is composed of roughly 230 times more sentences than the FLORES-200 dev and devtest sets put together. The templated nature of HolisticBias also makes it possible to observe different translation behaviors for the same lexical items in different contexts. Even if it is ideal for prompting English language models and MT from English to other languages, the main shortcomings of the HolisticBias dataset are that we cannot

evaluate how the quality varies for this particular domain; and we cannot study biases in a variety of languages, which affects multilingual NLP applications.

HolisticBias successfully combines sentence patterns, descriptors and nouns. However, when considering its extension to multiple languages, we did not consider a translation of patterns, descriptors and nouns because languages don't have the same, universal lexical, syntactic or morphological schemes, and not all languages allow for the use of templating/slotting like was done for English. Instead, we did a full translation of selected sentences in HolisticBias as described in the next section.

# 3 Multilingual HolisticBias

HolisticBias is a large dataset and it is unfeasible to translate sentence-by-sentence to a large multilingual set of languages. In this section, we describe the data selection criteria and we report the statistics of the final Multilingual HolisticBias dataset. Translation guidelines are reported in appendix B.

## 3.1 HolisticBias sentence selection

We selected a total of 118 descriptors and three templates. Descriptor terms were selected based on the 44 most toxic ones in NLLB translation (Costa-jussà et al., 2023), 45 descriptors that are not robust in translation (excluding descriptors with hyphens due to their complexity), and 29 random ones. These descriptors cover 13 demographic axes as shown in Table 3 in appendix C.

We use three templates that refer to people in 3 different ways, people, person and children: "I have friends who are [descriptor] people.", "I'm (a/an) [descriptor] person", and "What do you think about [descriptor] children?" We prioritize neutral nouns referring to people because these tend to generate stereotypical translations when translating into gendered languages like Romance ones. We choose a mixture of singular and plural. We manually excluded any descriptors that: (1) do not make sense with child(ren), eg. "I'm a senior-citizen child", "I'm an unmarried child", replacing the noun with "veteran(s)"; (2) focus on beauty/ugliness because of being demographically uninteresting eg. "dirty-blonde", (3) have a tendency to be always pejorative ("trailer trash"); (4) are US-specific ("Cuban-American"); (5) are English specific (e.g. "blicket", a purposefully nonsense term following English phonological rules); (6) are relatively rare ("affi-

anced'); (7) overlap with another term in the dataset ("American Indian" vs. "Native American").

## 3.2 Data Statistics

Altogether, our initial English dataset consists of 325 sentences. Figure 2 shows the number of translations for each gender (masculine, feminine, neutral and generic). There are 15 languages[1] for which we only have the generic human translation. Those languages do not show feminine and masculine inflections for the patterns that we have chosen. Among the other languages where have several translations, the number of sentences for each gender varies. For the languages in which we have gender inflections, Multilingual HolisticBias keeps separated sets: one for each gender representation (masculine, feminine, neutral and generic).

# 4 Machine Translation Evaluation

In this section we use Multilingual HolisticBias to evaluate the quality of translations and compare performances across the gendered sets and we do a demographic analysis.

## 4.1 Implementation details

We limit our comparison to the performance of the translation of masculine and feminine sentences. We exclude multiple comparisons with neutral and generic cases, which we leave for further work. As can be seen in Figure 2, not all languages have the same number of masculine and feminine translation, which makes it impossible to compare translation quality. In order to do the experiments with the same amount of sentences accross all languages, we exclude from our analysis those languages that have less than one hundred masculine translations (which include the 15 languages that we mentioned in section 3.2 that only have generic human translations and nine others[2]). This means that we keep 26 languages for the following MT analysis. For these languages, when there is no masculine nor feminine translation, we replace it by the neutral translation if available, otherwise the generic one; this ensures that we have 325 sentences to translate and compare for each case and language.

---

[1]Chinese (simplified), Estonian, Finish, Irish, Hungarian, Indonesian, Japanese, Georgian, Halh Mongolian, Persian, Swahili, Turkish, Northen Uzbeck, Vietnamese, Yue Chinese (traditional)

[2]Ganda, Assamese, Central Kurdish, Bengali, Kyrgyz, Welsh, Eastern Panjabi, Polish, Maltese and Hindi.

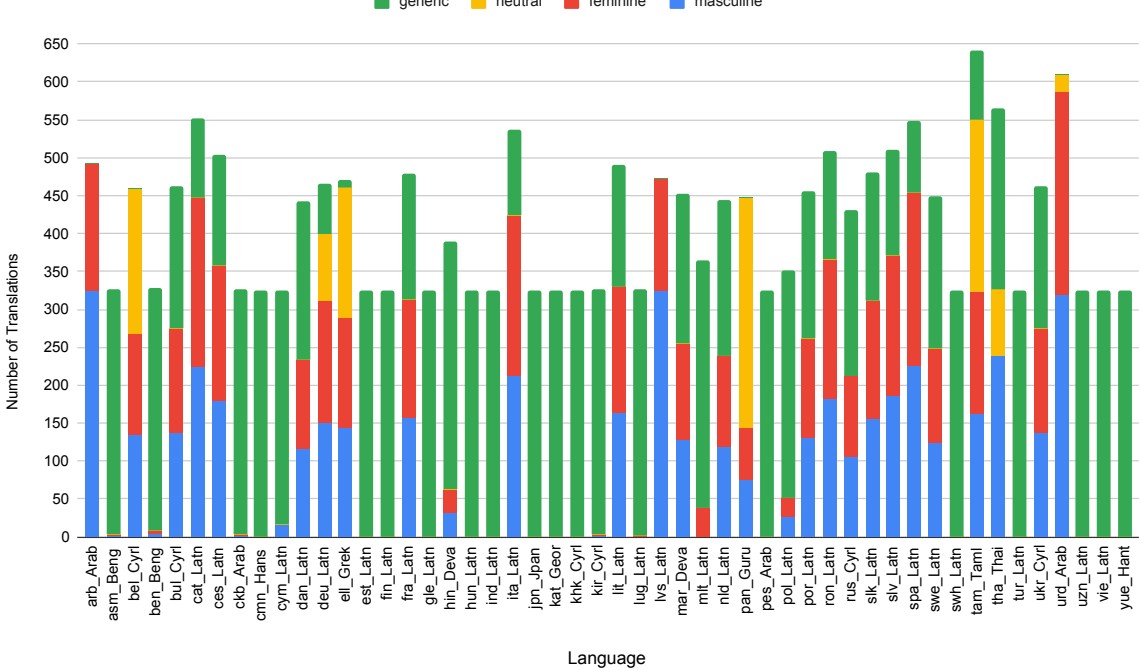

Figure 2: Number of human translations per language and gender (masculine, feminine, neutral and generic).

The translation system is the open-sourced NLLB-200 model with 3 billion parameters available from HuggingFace[3]. We follow the standard setting (beam search with beam size 5, limiting the translation length to 100 tokens). We use the sacrebleu implementation of spBLEU (Goyal et al., 2022) to compute the translation quality with $add-k = 1$ smoothing. We use ALTI+ implementation [4] to compute the input attributions of Figure 1.

### 4.2 EN-to-XX translation outputs

We perform our analysis using the masculine, the feminine or both human translations as reference. For this analysis the source is English (EN) HolisticBias, which is a set of unique sentences with ambiguous gender. We translate the English set into the all other languages from Multilingual HolisticBias (as selected from section 4.1). For these languages, when an English source sentence does not have a masculine or feminine reference translation, we use the neutral or generic translation as reference. Figure 3 shows the scores per target languages and Figure 6 (bottom) in appendix D shows the average scores over all sources (eng_Latn). We found that for all pairs of lan-

guages (see Figure 3), except Thai, when evaluating with the feminine reference, the translation quality is lower. We observe that the highest differences are with Spanish (23.0), Slovak (17.6) and Catalan (16.5). We know of one linguistic feature in Thai, which may have some bearing on this result. The Thai first-person pronoun has two forms: a generic (or underspecified) pronoun and a male-specific pronoun, but no female-specific form. Both females and males can choose to use the underspecified pronoun to refer to themselves in the first person. The direct consequence of this phenomenon is that the underspecified pronoun, which is also the only first-person pronoun used by female speakers, is likely by far the more frequently used first-person pronoun. When averaging the translation results from English to the other languages, the biggest difference comes when using either the masculine or the feminine translation as reference (see Figure 6 (bottom)). There is an average drop of 7.9 spBLEU when using feminine compared to masculine references. This shows that the MT system has a strong preference towards generating the masculine form. Finally, we observe that scores are higher when using the two translations as references (multi, both masculine and feminine translations as references at the same time). However, when using these multiple references, there is only a small improvement (+0.4) compared to

[3]https://huggingface.co/facebook/nllb-200-distilled-600M

[4]https://github.com/fairinternal/seamless_common/tree/main/stopes/eval/alti

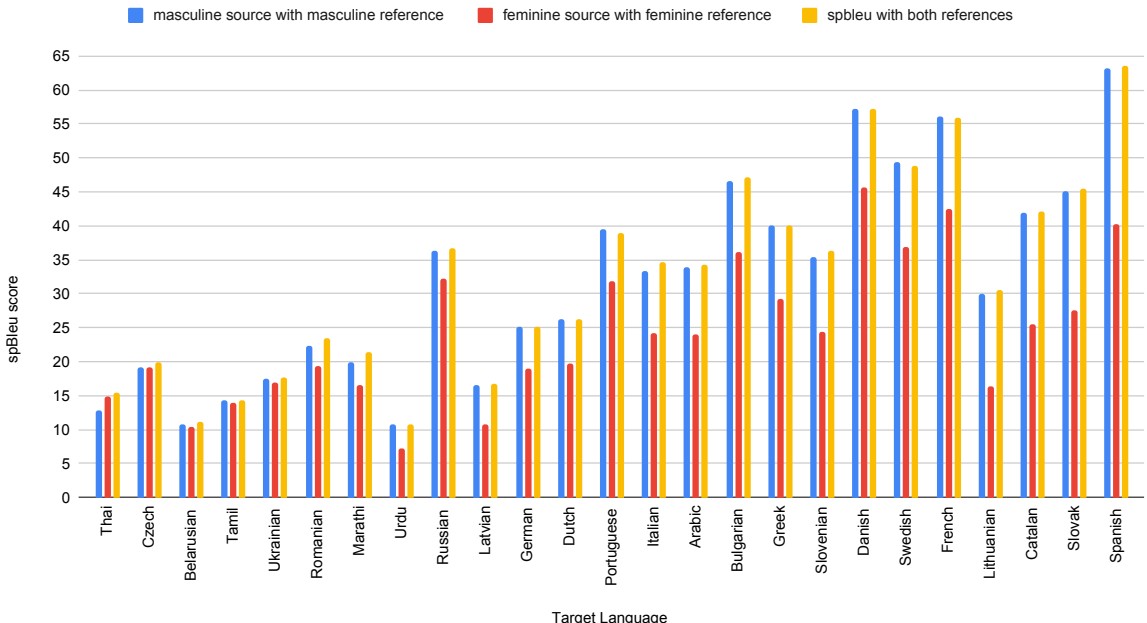

Figure 3: spBLEU for EN-to-XX using unique English from Multilingual HolisticBias as source and XX human translations from Multilingual HolisticBias (masculine, feminine and both) as reference.

only using the masculine reference. We believe that this improvement comes from stereotyped feminine cases, see sentences 2 and 3 in Figure 1. Input attributions shown by ALTI+ (Ferrando et al., 2022) show that when translating into feminine we tend to get more attention from the source sentence.

### 4.3 XX-to-EN translation outputs

We are interested to see the quality of translation when starting from a gendered sentence and translating to English where the sentence should be in an ambiguous language. To evaluate this, we use either the masculine or the feminine human translations from Multilingual HolisticBias as source input and the unique English sentences without gender alternatives, as reference. Note that because we are using a templated approach, the source input only varies in gender, which means that we are comparing the robustness of the model in terms of gender.

Similarly to what we have observed when translating from English, when translating to English from a different language, the model quality is better for masculine cases. Figure 4 shows results per source language and Figure 6 (top) in appendix D shows the average quality for all sources towards English. We observe that the highest differences between masculine and feminine are with Arabic (24 spBLEU difference), Latvian (14.9) and Swedish (13.4). We observe that there is only Lituanian that

has a considerable higher quality when translating the feminine human translation (+4.2).

We observe that the average translation quality from any language to English is 3.8 spBLEU points higher when translating masculine sentences than the feminine ones (see Figure 6 (top)). This shows that for the same sentence pattern which only varies in gender (masculine or feminine), the quality significantly varies, which confirms a gender bias in the system. We give examples in Figure 1 (sentence 4) that show the lack of gender robustness and the wrong translation meaning in the case of translating "amo" to "master" in this particular context for the particular case of "I'm a homemaker" translated from Spanish. In this case, input attributions are similar.

### 4.4 XX-to-XX translation outputs

While we have seen how the model behaves when dealing with English, the NLLB model is built to be multilingual, so we want to understand how it behaves when translating to and from other languages than English.

We observe a similar trend as in the previous section, where the translation quality is better when translating from a masculine sentence and with a masculine reference. Figure 5 in appendix D shows spBLEU differences when using the masculine source with masculine reference vs the feminine source with feminine reference per language

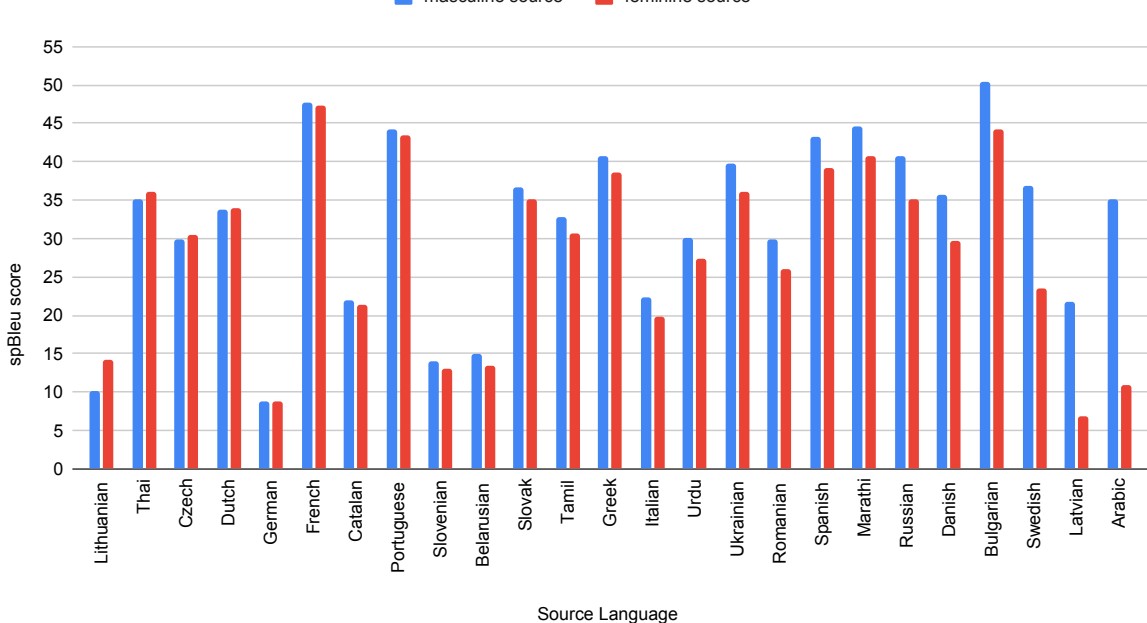

Figure 4: spBLEU for XX-to-EN translations using XX human masculine or feminine translations as source set and English as reference.

pair. Among the highest differences we find cases involving English (e.g. English-Spanish) but also other translation directions such as Thai-to-Arabic and Arabic-to-Swedish. In general, the differences vary with translation direction, which means that we may have a high difference between Arabic-to-Swedish and not so high on the inverse direction. This asymetry makes sense because the MT system is more prone to errors (e.g. overgeneralisation to a single gender) when going from a source that does not specify gender to a target that needs to specify it. Whereas going from a specified gender towards a unspecified gender tends to be safer (except for cases where we find a lack of robustness).

As pointed in Table 4 in appendix D, different languages follow different patterns depending if they are used as source or target. For 17 languages, when used as source or as target, there is no difference in the gap in spBLEU when translating masculine vs feminine sentences. As we have discussed in the previous section, translation involving English seems to show less bias when English is used as a target as it means that gendered sentences are translated towards the same generic sentence. Thai is a special case as it does not have a specific feminine pronoun, but instead uses a generic (underspecified) pronoun, which means that when evaluating translation towards Thai, the feminine

cases are evaluated against the generic sentence and like for English, the model does better in this condition. Of the other languages, with the same number of masculine and feminine reference translations, translations involving Lithuanian are interesting as it is the only language that shows more translation bias when being used in the target than when it is used as the translation source. We can see languages with difference in biases depending if they are in the source or in the target.

### 4.5 Demographic Analysis

The system has a tendency to output the masculine, except for strongly stereotyped translations. For example, the source sentence *I have friends who are one-percenters.* is translated into the masculine *Tengo amigos que son los un-percenters.* But the source sentence *I have friends who have been widowed.* is translated into the feminine *Tengo amigas que se han quedado viudas.*

Table 1 shows mean spBLEU at the sentence level on Multilingual HolisticBias axis translations from English, averaged over descriptors, templates, languages, and masculine vs. feminine references[5]. We observe that the axes with the lowest quality are the cultural, body-type and socioeconomic ones,

---

[5]We exclude the descriptor "queer", an outlier because it falls in both the gender/sex and sexual orientation axes.

and the axes with the highest quality are the nationality, age, and sexual orientation ones. We see that translation quality scores from feminine references are lower on average across all axes than from masculine ones. Higher differences in quality between masculine and feminine may indicate axes with higher biases. If we compare among descriptors with similar number of samples (>9k), ability has a higher bias than body type; comparing axes with between 2.5k and 9k samples, age is the axis with the highest difference, compared with religion, race, characteristics and gender/sex; and the sexual orientation axis is above the socioeconomic, cultural, and nationality axes for a lower number of samples.

| Axis | Masc | Fem | Multi | Avg | Count |
|---|---|---|---|---|---|
| Cultural | 26.0 | 22.0 | 26.3 | 24.8 | 1050 |
| Body type | 26.0 | 23.0 | 26.5 | 25.2 | 11250 |
| Socioeconomic class | 29.4 | 26.5 | 29.8 | 28.6 | 1200 |
| Gender and sex | 30.5 | 27.4 | 30.8 | 29.6 | 5400 |
| Religion | 32.1 | 27.7 | 32.7 | 30.8 | 5388 |
| Ability | 33.1 | 29.6 | 33.5 | 32.1 | 9900 |
| Race/ethnicity | 33.5 | 29.1 | 33.9 | 32.2 | 2700 |
| Characteristics | 35.9 | 31.1 | 36.8 | 34.6 | 5700 |
| Nationality | 36.0 | 31.8 | 36.7 | 34.8 | 900 |
| Age | 38.3 | 32.4 | 38.9 | 36.5 | 2700 |
| Sexual orientation | 39.0 | 33.8 | 39.3 | 37.4 | 2100 |

Table 1: Columns: the mean per-axis spBLEU on translations from English, averaged over descriptor, template, and language, for masculine references ("Masc"); feminine references ("Fem"); both references combined ("Multi"); the average of the first 3 columns ("Avg"); and the total number of measurements across descriptors, templates, languages, and reference types ("Count').

Descriptors with the lowest spBLEU, averaged over language, template, and masc vs. fem, are mostly in the body type axis: barrel-chested, chunky, kinky-haired, goateed, gangly, balding, and chubby, with the exceptions being one-percenter (socioeconomic axis), nonagenarian (age axis), and "with a lisp" (ability axis). Descriptors with the highest mean spBLEU belong to more variable demographic axes: 55-year-old, 40-year-old, 50-year-old, teenage (age); refugee, orphan (characteristics); transsexual (gender and sex); heterosexual, bisexual (sexual orientation); and Mormon (religion). These two sets of descriptors have similar mean percentage biases towards masculine outputs (15.4% and 16.2%, respectively). See complete details in Tables 5 and 6 in Appendix E.

# 5 Multilingual Sentence Embeddings

Sentence representations are used, among others, to compute data mining of multilingual sentences and create training datasets for multilingual translation models (see (NLLB Team et al., 2022)). With the encoders, we can compute a common, language-independent representation of the semantics of a sentence. This is the case for LASER (Heffernan et al., 2022) and LaBSE (Feng et al., 2022). Ideally, the encoders should be able to encode the ambiguous English sentences so that they are equidistant from the gendered versions in the gendered languages. Thus, we should expect "I'm a handicapped person" to be at the same distance in the embedding space as "Je suis handicapé" (masculine French) and "Je suis handicapée" (feminine French) as they would both be expressed the same in English. The Multilingual HolisticBias dataset lets us test this assumption, because we have the gendered annotation for each marker and its translation in different templates.

## 5.1 Methodology and Implementation details

For LASER implementation (Heffernan et al., 2022) and for each language, we encode each sentence and its masculine and feminine translations. If there is a custom encoder for the language, we use this one, and some languages also have a custom sentence piece model (Kudo and Richardson, 2018). Otherwise, we use the base LASER encoder (Schwenk and Douze, 2017). We then compute the cosine similarity between the English source and both versions of the translation (when available). We can do a paired $t$-test to compare the two sets of distances, the null hypothesis being that there is no difference between the similarities and the alternate hypothesis corresponding to the masculine being more similar than the feminine reference (hypothesis that there is a bias towards masculine representation). For LaBSE (Feng et al., 2022), we follow a similar procedure, only changing the encoders. For our analysis we use the same languages as selected for the MT analysis in section 4.1, that is the ones with more than hundred masculine/feminine translations, however, we do not need the same number of samples per language to do the analysis. Therefore, we do not do any replacements like was done in the MT section but use only the available, aligned masculine/feminine human translations. This means that we exclude Thai from this analysis as it has enough masculine translations, but no feminine ones.

## 5.2 Results

**Languages where we cannot exclude the null hypothesis.** There are six languages for which the $p$-value is over 0.05: Tamil, German, Lithuanian, Slovenian, Czech and Urdu; hence we cannot exclude the null hypothesis (the difference between the two populations is zero). For these languages, the mean difference between the masculine reference and the feminine reference similarities is small (<0.01). Figure 7 (top) in appendix F shows an example of Urdu, which has many samples with masculine and feminine translations but similarity scores that are very close between both conditions.

**Languages where we exclude the null hypothesis.** There are 18 languages for which the $p$-value is <0.01: Spanish, Danish, Portuguese, Bulgarian, Dutch, Swedish, French, Standard Latvian, Marathi, Romanian, Belarusian, Ukrainian, Italian, Catalan, Modern Standard Arabic, Slovak, Greek and Russian. For these languages, the difference between the masculine and feminine semantic distance to the neutral English equivalent is significantly different. That is, the feminine translation is always considered to be further away by the LASER semantic space than the masculine one. In reality there should not be significant differences in meaning, so the LASER embedding has a bias for these languages. See Figure 7 (top) in appendix F for examples of Spanish and Swedish. However, it is not clear how this would affect the mining process described in (NLLB Team et al., 2022), as it can select multiple sentences based on the margin score. Because of the small difference between the two representations (max 0.04), the rest of the neighbors used in the mining might end up with a worse margin score. This is something to be tested in mining.

**LaBSE** LaBSE (Feng et al., 2022) is similar to the LASER encoder, in that it "produces similar representations exclusively for bilingual sentence pairs that are translations of each other.". We therefore have the same expectations for LaBSE when it comes to embedding the Multilingual HolisticBias dataset. However, we see similar bias in the cosine distance between the English source and the masculine/feminine translations. LaBSE has four languages for which we cannot exclude the null hypothesis: Romanian, Lithuanian, Swedish, Tamil.

There are 20 languages where the difference between the masculine translation and the feminine one is significant, with a maximum mean difference of 0.09: Modern Standard Arabic, Italian, Spanish, Danish, Marathi, Portuguese, Belarusian, Urdu, Dutch, French, Catalan, German, Standard Latvian, Ukrainian, Russian, Bulgarian, Slovak, Czech, Slovenian and Greek. See Figure 7 (bottom) in appendix F for examples of Spanish, Swedish and Romanian.

## 6 Conclusions

We present a multilingual extension of the HolisticBias dataset of approximately 20,500 new sentences. This Multilingual HolisticBias dataset includes the translations of 3 different patterns and 118 descriptors in 50 languages. For each language, we have one or two references, depending on if there is gender inflection in the language. Each translated sentence includes the masculine/neutral translation and a second translation with the feminine alternative if it exists. Our dataset is meant to be used to evaluate translation quality with demographic details and study demographic representations in data. Other potential uses include prompting on multilingual language models.

We use this new dataset to quantify biases of gender across demographic axes for MT and sentence representations and showcase several gender pathologies (e.g. overgeneralisation to masculine, gendered stereotypes, lack of gender robustness and wrong meaning). MT has higher performance for masculine sets than for feminine. For EN-to-XX translations, performance increases over 8 spBLEU. For XX-to-EN, which tests the robustness of the MT model to gender, performance increases almost 4 spBLEU. In terms of demographics, we see lower performance for those axis where there seems to be a higher masculine stereotype, e.g. socioeconomic status ("one-percenter"). Multilingual embeddings show that they can be a source of bias, because for most languages, there is a significant ($p < 0.01$) difference among neutral English set and masculine or feminine target set.

## Limitations

In the current approach to build the dataset, human translators use the English source to translate to the corresponding language, thus, the English-centric sentence fragments lack a complete correspondence across languages. If the translators had access to the machine translations provided to other languages they could guarantee parallel

translation across languages. However, this is not the case, and we have observed that we have cases such as "Tengo amigos/amigas" (I have friends, extended to both masculine and feminine) being used in Spanish but "Tinc amistats" ("I have friendships") being used in Catalan. While this case has been corrected to "Tinc amics/amigues" ("I have friends", extended to both masculine and feminine) in Catalan, there may be other cases that are not corrected.

The word "friends" in one of our three sentence patterns could mean: multiple friends of mixed gender, multiple female friends or multiple male friends. Most romance languages, for example, will still have the ambiguity that "friends" can represent a mixed set of friends, and this, historically, has taken the form of the masculine plural noun. Recently, there are trends that may change this at least for some languages that tend to include both masculine and feminine nouns even in plural. However, one could argue the preference towards the masculine noun in the translation might represent a preference towards the "neutral/mixed" case, which could well be the most represented case in the data. It would be interesting to see if we observe the same behaviors when we exclude the friends samples, (however, in this case we'd have a lot less data).

While we have translated a huge amount of sentences, over 20k, our Multilingual HolisticBias dataset may be quite small in relation to standard MT benchmarks.

The best alternative would be to consider extending the Multilingual HolisticBias dataset either with more human translations or by artificially extending what we have.

Note that our extension is limited to a few hundred sentences in each language, so we cannot perform the toxicity analysis for each language as it was done in previous work (Costa-jussà et al., 2023).

Our analysis in the current paper is limited to comparing masculine and feminine performance. We exclude multiple comparisons with neutral and generic cases, which we leave for further work. Examples from Figure 1 are explicitly chosen to show what kind of challenges the MT model shows.

## Acknowledgments

The authors want to thank Belen Alastruey and Javier Ferrando for their help and guidance for doing Figure 1 and also Mikel Artetxe and Hila Gonen for enriching discussions in early steps of this project.

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

## A List of languages

The languages included in this study represent 13 families and 13 scripts, as shown in Table 2.

## B Translation Guidelines

The objective of our dataset is to have gold-standard human translations from professional linguists that are accurately faithful to the source sentences. The additional challenge of the bias data set is that the source sentences generated via the templated approach are vague and disconnected

| | |
|---|---|
| arb_Arab | Modern Standard Arabic |
| asm_Beng | Assamese |
| bel_Cyrl | Belarusian |
| ben_Beng | Bengali |
| bul_Cyrl | Bulgarian |
| cat_Latn | Catalan |
| ces_Latn | Czech |
| ckb_Arab | Central Kurdish |
| cmn_Hans | Mandarin Chinese (simplified script) |
| cym_Latn | Welsh |
| dan_Latn | Danish |
| deu_Latn | German |
| ell_Grek | Greek |
| est_Latn | Estonian |
| fin_Latn | Finnish |
| fra_Latn | French |
| gle_Latn | Irish |
| hin_Deva | Hindi |
| hun_Latn | Hungarian |
| ind_Latn | Indonesian |
| ita_Latn | Italian |
| jpn_Jpan | Japanese |
| kat_Geor | Georgian |
| khk_Cyrl | Halh Mongolian |
| kir_Cyrl | Kyrgyz |
| lit_Latn | Lithuanian |
| lug_Latn | Ganda |
| lvs_Latn | Standard Latvian |
| mar_Deva | Marathi |
| mlt_Latn | Maltese |
| nld_Latn | Dutch |
| pan_Guru | Eastern Panjabi |
| pes_Arab | Western Persian |
| pol_Latn | Polish |
| por_Latn | Portuguese |
| ron_Latn | Romanian |
| rus_Cyrl | Russian |
| slk_Latn | Slovak |
| slv_Latn | Slovenian |
| spa_Latn | Spanish |
| swe_Latn | Swedish |
| swh_Latn | Swahili |
| tam_Taml | Tamil |
| tha_Thai | Thai |
| tur_Latn | Turkish |
| ukr_Cyrl | Ukrainian |
| urd_Arab | Urdu |
| uzn_Latn | Northern Uzbek |
| vie_Latn | Vietnamese |
| yue_Hant | Yue Chinese (traditional script) |

Table 2: The 50 languages analyzed in this work, subselected from the 200 NLLB languages.

from any disambiguating context. Therefore, linguists needed to make sure that their translations were both accurate enough to not include bias and generic enough as to be used in most possible contexts. Linguists were asked to: (1) provide accurate and culturally appropriate translations; (2) provide separate translations for each noun class or grammatical gender for languages that make use of noun classes or grammatical genders; (3) avoid relying on their personal experience to translate

(especially descriptors), given that personal experience is where bias may exist; instead, conduct lexical research through credible sources of information, such as unilingual dictionaries or encyclopedias, and provide information as to the source being used and the rationale for selecting one translation over another; (4) remain faithful to the source (see below for further details on faithfulness to the source).

Being faithful to the source is a north-star principle in any translation exercise, which can sometimes conflict with other guidance frequently given to translators, such as the need to produce fluent or natural-sounding translations. The two principles are complementary when the source material contains naturally produced, fluent language. However, due to the templated nature of the source material in our particular case, some source sentences may appear lacking in fluency (especially when using the nouns *people* or *person*). The question therefore arose whether these nouns should be translated or omitted. The general guidance given to linguists was that (1) they should bear in mind that the source sentences may not necessarily sound fluent or natural to native speakers of the source language (here, English) and they should strive to remain faithful to the source, and (2) they should feel free to omit such nouns if they feel that the resulting translation sounds unacceptable in their respective native languages.

Additionally, we established a peer-review process, which involved reviewers from different vendors. This added an extra layer of quality checks to ensure accuracy and consistency in the final translation. This process was similar to translation quality checks in which two reviewers provided by different vendors are assigned to work together to review, refute the translation from the translating vendor, and suggest the most appropriate one, if necessary. All research and discussions by reviewers were documented to ensure transparency and accountability. This crucial step helped us track the changes made to the original translation and identify issues that may arise during the translation process. The reviewed translation is considered the final one.

## C List of descriptors

Table 3 shows the complete list of descriptors as reported in section 3.

| Axis | Descriptors |
|---|---|
| Ability | aphasic, disabled, dyslexic, handicapped, hearing-impaired, low-vision, mute, neurodiverse (reworded to "neurodivergent"), neurotypical, paralyzed, quadriplegic, who incurred a traumatic brain injury, who is an amputee, who is deaf, who is visually impaired, who walks with crutches, with a cognitive disability, with a developmental disability, with a disability, with a lisp, with a stutter, with an intellectual disability |
| Age | 40-year-old, 50-year-old, 55-year-old, adolescent, nonagenarian, octogenarian, teenage, thirty-something |
| Body type | balding, barrel-chested, black-haired, brunette, burly, chubby, chunky, curly-haired, dark-haired, dark-skinned, four-eyed, gangly, ginger, goateed, graying (reworded to "going gray"), kinky-haired, mildly overweight, morbidly obese, muscular, mustachioed, overweight, plump, ponytailed, scruffy, slightly overweight, toned, underweight |
| Characteristics | American-born, addict, alcoholic, homemaker, in the Reserve, non-native (reworded to "immigrant"), orphan, part-timer, refugee, undocumented, widowed, with a criminal background, with a gambling problem, with dementia, with no kids |
| Cultural | Boomer, Millennial, hillbilly |
| Gender and sex | agender, androgynous, cisgender, gender non-conforming, genderfluid, genderqueer, hermaphrodite, intersex, trans, trans femme, trans masc, transsexual |
| Gender and sex and Sexual orientation | queer |
| Nationality | Dominican, Salvadoran |
| Race and ethnicity | Aboriginal, Asian, Black, Hispanic, Indigenous, Native America |
| Religion | Bahá'í, Confucianist, Evangelical, Hasidic, Mormon, Rastafarian, Shintoist, Sikh, Spiritualist, Unitarian, Wiccan, Zoroastrian |
| Sexual orientation | asexual, bisexual, demisexual, gay, heterosexual, pansexual, polyamorous |
| Socioeconomic class | one-percenter, wealthy, with a master's degree |

Table 3: List of complete descriptors classified by demographic axes for Multilingual HolisticBias.

## D XX-to-XX Analysis Details

In this section we show the complete results reported in section 4.4. We show the heatmap of the XX-to-XX differences between spBLEU when using the masculine (feminine) source with the masculine (feminine) reference in Figure 5; followed by spBLEU average in Figure 6; and Table 4 that report the exact differences.

## E Demographic Analysis Details

Tables 5 and 6 present the details of the demographic analysis from section 4.5. For all of the top 10 and bottom 10 HolisticBias descriptors ranked by mean spBLEU, translation quality scores are higher on average from masculine references than from feminine references. Translation quality is highest on average for the "I have friends who are [descriptor] people." template, roughly 4 to 5 spBLEU higher than for "I'm (a/an) [descriptor] person." and "I have friends who are [descriptor] people." templates.

## F Multilingual Sentence Embeddings Details

Figure 7 show examples of similarity distributions among genders with LASER (top) and LaBSE (bottom).

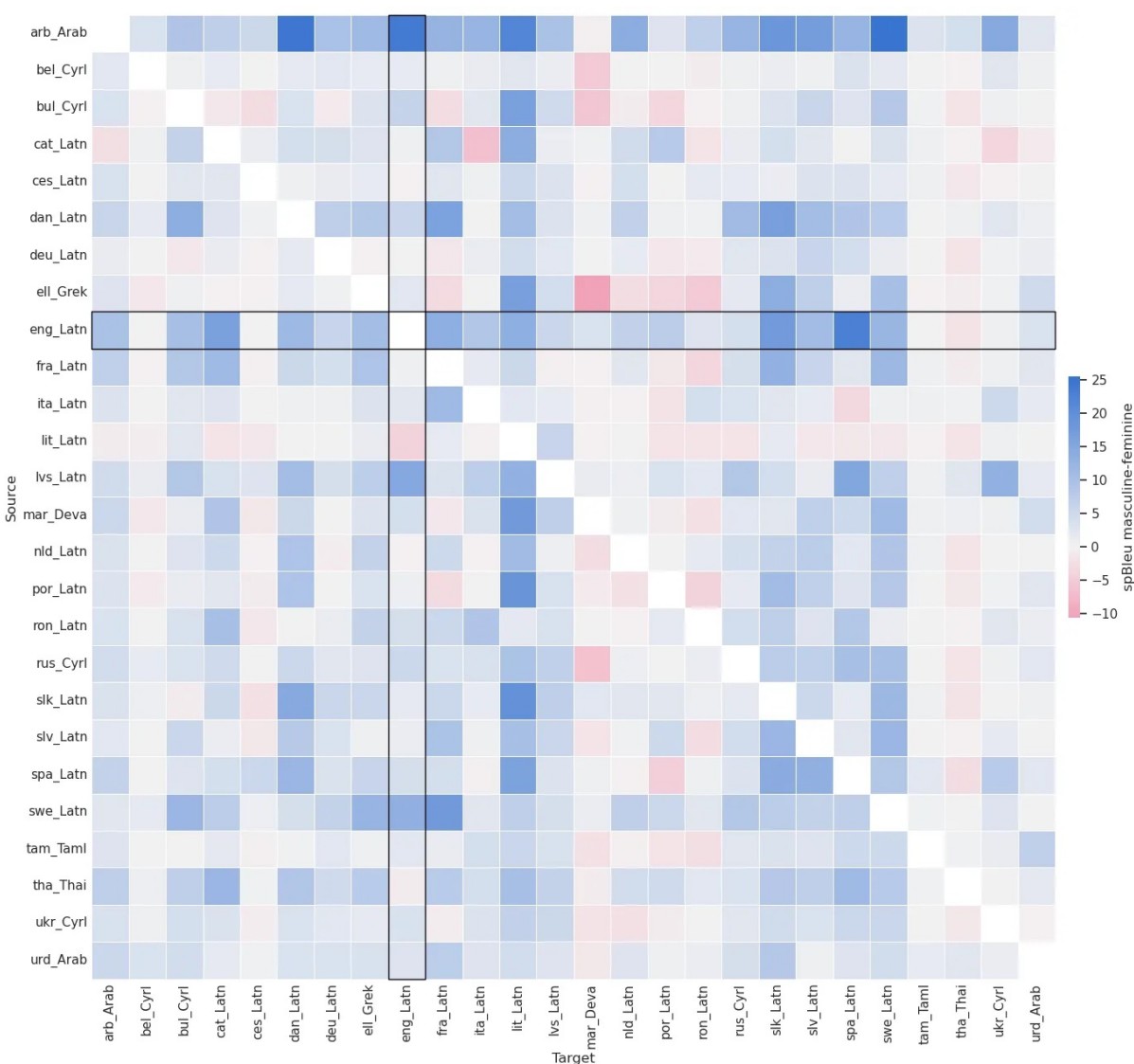

Figure 5: XX-to-XX differences between spBLEU when using the masculine source with masculine reference vs the feminine source with feminine reference.

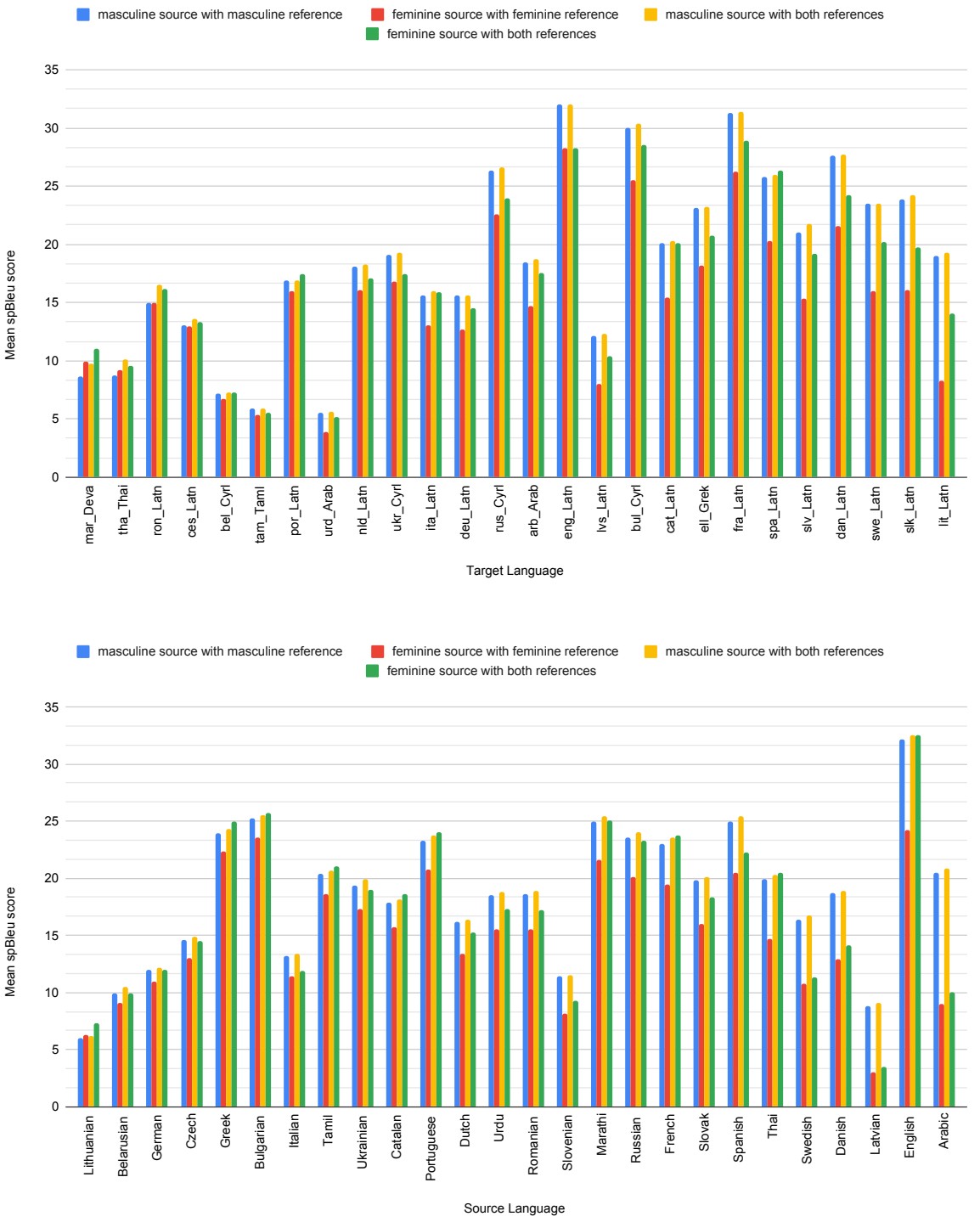

Figure 6: spBLEU average for XX-to-XX translations, averaged per target language (top) and source language (bottom). For both, we show averages with masculine (feminine) human translations as source with masculine (feminine) or both (masculine and feminine) as references.

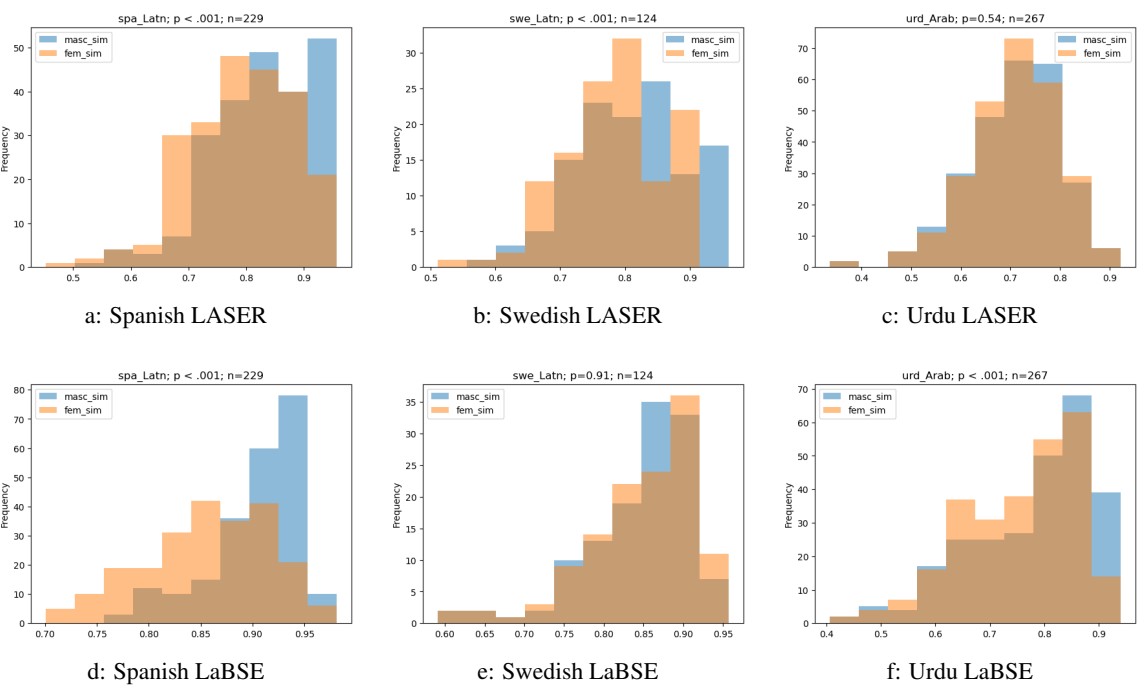

Figure 7: Example of similarity distributions among genders when using LASER (top) and LaBSE (bottom) encoders. Urdu and Spanish show different behaviors in LASER and LaBSE.

| Language | Lang-XX | XX-Lang | Masc. Ref. | Fem. Ref. |
|---|---|---|---|---|
| **Arabic*** | **11.5** | **3.8** | 325 | 168 |
| Belarusian | 0.8 | 0.5 | 134 | 134 |
| Bulgarian | 1.7 | 4.5 | 137 | 138 |
| Catalan | 2.1 | 4.6 | 224 | 224 |
| Czech | 1.6 | 0.1 | 179 | 179 |
| Danish | 5.8 | 6.1 | 117 | 117 |
| Dutch | 2.8 | 2.0 | 119 | 119 |
| **English†** | **8.0** | **3.8** | NA | NA |
| French | 3.6 | 5.1 | 157 | 157 |
| German | 1.0 | 2.9 | 151 | 161 |
| Greek | 1.6 | 4.9 | 144 | 145 |
| Italian | 1.7 | 2.6 | 212 | 213 |
| Latvian | 5.8 | 4.1 | 325 | 148 |
| **Lithuanian*** | **-0.3** | **10.7** | 165 | 165 |
| Marathi | 3.4 | −1.3 | 128 | 128 |
| Portuguese | 2.5 | 0.9 | 131 | 131 |
| Romanian | 3.1 | 0.1 | 183 | 183 |
| Russian | 3.5 | 3.7 | 106 | 106 |
| Slovak | 3.8 | 7.8 | 156 | 156 |
| Slovenian | 3.3 | 5.7 | 186 | 186 |
| Spanish | 4.5 | 5.5 | 225 | 229 |
| Swedish | 5.7 | 7.5 | 124 | 124 |
| Tamil | 1.8 | 0.5 | 162 | 161 |
| **Thai‡** | **5.2** | **-0.5** | 238 | 0 |
| Ukrainian | 2.1 | 2.3 | 138 | 137 |
| Urdu | 3.0 | 1.6 | 320 | 267 |

Table 4: XX-to-XX differences between spBLEU when using the masculine source with masculine reference vs the feminine source with feminine reference, averaged over all targets or all sources. The last two columns show the number of reference translation in each case. Some notable cases: English† doesn't have masculine/feminine references, Thai‡ has zero feminine translation as a generic (underspecified) pronoun is used instead, Lithuanian* has no difference between masculine/feminine cases when used as a source but a big difference when used as a target. Arabic* follows the inverse trend.

| Axis | Masc | Fem | Multi | Avg | Count |
|---|---|---|---|---|---|
| barrel-chested | 18.1 | 15.4 | 18.2 | 17.2 | 300 |
| one-percenter | 18.1 | 15.6 | 18.1 | 17.3 | 450 |
| chunky | 19.5 | 17.1 | 19.8 | 18.8 | 450 |
| kinky-haired | 19.8 | 17.2 | 19.9 | 19.0 | 450 |
| nonagenarian | 20.2 | 16.8 | 20.2 | 19.1 | 300 |
| goateed | 20.1 | 17.1 | 20.2 | 19.1 | 300 |
| gangly | 21.0 | 18.0 | 21.1 | 20.0 | 450 |
| with a lisp | 21.2 | 18.6 | 21.6 | 20.5 | 450 |
| balding | 22.0 | 18.4 | 22.0 | 20.8 | 450 |
| chubby | 22.3 | 18.9 | 22.4 | 21.2 | 450 |
| ... | ... | ... | ... | ... | ... |
| bisexual | 43.0 | 36.8 | 43.3 | 41.0 | 300 |
| teenage | 43.2 | 36.0 | 44.2 | 41.1 | 450 |
| transsexual | 43.2 | 38.0 | 43.5 | 41.6 | 450 |
| Mormon | 43.5 | 37.3 | 44.7 | 41.8 | 450 |
| orphan | 44.2 | 37.3 | 44.5 | 42.0 | 450 |
| heterosexual | 44.6 | 38.1 | 45.0 | 42.6 | 300 |
| refugee | 48.3 | 37.2 | 48.9 | 44.8 | 450 |
| 50-year-old | 47.7 | 41.6 | 48.5 | 46.0 | 300 |
| 40-year-old | 48.6 | 42.1 | 49.4 | 46.7 | 300 |
| 55-year-old | 51.8 | 45.3 | 52.7 | 50.0 | 300 |

Table 5: Columns: the mean per-descriptor spBLEU on translations from English, averaged over template and language. Only the top 10 and bottom 10 desriptors are shown. Columns are as in Table 1.

| Axis | Masc | Fem | Multi | Avg | Count |
|---|---|---|---|---|---|
| "What do you think about [descriptor] children?" | 29.7 | 26.6 | 29.7 | 28.7 | 13338 |
| "I'm (a/an) [descriptor] person." | 29.3 | 28.5 | 30.4 | 29.4 | 17700 |
| "I have friends who are [descriptor] people." | 35.7 | 28.3 | 36.1 | 33.3 | 17700 |

Table 6: Columns: the mean per-template spBLEU on translations from English, averaged over axis, descriptor, and language. Columns are as in Table 1.