# OpenReview forum: "Multilingual Holistic Bias: Extending Descriptors and Patterns to Unveil Demographic Biases in Languages at Scale"
_EMNLP/2023/Conference — EMNLP 2023 Main_

### Official Review · Reviewer_seQ7 · 2023-08-01

**Soundness:** 3

**Excitement:**

4: Strong: This paper deepens the understanding of some phenomenon or lowers the barriers to an existing research direction.

**Paper Topic And Main Contributions:**

This paper aims to measure demographic bias in multilingual machine translation systems, e.g., NLLB, and multilingual sentence encoders, e.g., LASER and LaBSE, in a variety of languages.

Firstly, it extends the English-only bias measurement dataset, HolisticBias, to MultilingualHolisticBias covering 50 typologically-diverse languages with professional linguists, translation guidelines, and quality check. The dataset has 20k sentences in total and covers 3 demographic axes.

Using MultilingualHolisticBias, it conducts some comprehensive analysis on multilingual NLP applications. For multilingual translation models, results on NLLB shows that the system has a tendency to output the masculine, except for strongly stereotyped translations. For multilingual sentence encoders, this paper find that for most languages masculine translations are significantly closer to the English neutral sentences when embedded.

**Questions For The Authors:**

A. For the languages with only generic translations, e.g., Chinese, are these data useful for demographic bias measurement.

B. Why are only 3 templates selected? In addition to the ‘people, person and children’ reason, are there any other reasons for choosing these 3 templates?

C. What is add-k=1 smoothing?

**Reasons To Accept:**

- This paper proposes a multilingual bias measurement dataset covering 50 languages, which is valuable for measuring demographic biases in a variety of languages.
- This paper investigates the demographic biases in both multilingual machine translation systems and multilingual sentence encoders. The comprehensive results show that demographic biases exists in these multilingual models.

**Reasons To Reject:**

- For each language, the number of sentences is from 325 to 650 , which is limited and the results may be affected by a few samples.

**Reproducibility:**

3: Could reproduce the results with some difficulty. The settings of parameters are underspecified or subjectively determined; the training/evaluation data are not widely available.

**Reviewer Confidence:**

4: Quite sure. I tried to check the important points carefully. It's unlikely, though conceivable, that I missed something that should affect my ratings.

---

> ### Author Rebuttal · Authors · 2023-08-28
>
> Thanks for the comments!
>
> QA. Yes. we can study the quality of the translations in each demographic axis. However, results are not strictly comparable since different demographics use different vocabulary. But still, it is useful!
>
> QB. We limited to 3 patterns because of budget constraints. Translations are professionally created with strict quality checks.
>
> QC. This is the smoothing option that we choose available in SacreBleu. We add this detail for completeness.

---

### Official Review · Reviewer_QhKn · 2023-08-04

**Soundness:** 3

**Excitement:**

4: Strong: This paper deepens the understanding of some phenomenon or lowers the barriers to an existing research direction.

**Missing References:**

There has been some work on reporting gender bias in MT when translating from English to a gender-marking language, specifically Arabic. I find these papers to be relevant:

1) https://aclanthology.org/2022.naacl-main.46.pdf
2) https://aclanthology.org/W19-3822.pdf
3) https://aclanthology.org/2022.lrec-1.199.pdf


**Paper Topic And Main Contributions:**

In this paper, the authors introduce MultilingualHolisticBias, a multilingual extension of the HolisticBias dataset. HolisticBias is a dataset of 472K English sentences that are created through templates. These templates are filled with descriptors that cover various demographical axes such as race, gender, and race.

The author selected a subset of 325 sentences from HolisticBias bias which includes 118 demographic descriptors covering 13 demographic axes, and three templates. They manually introduced the target translations of the 325 sentences for 50 languages. The translations include gender-rewritten (or gender-neutral) alternatives for gender-marking languages when there's ambiguity in English. Therefore, each English sentence will have one or two references depending if the target language is gender-marking or not. In total, MultilingualHolisticBias consists of ~20.5K multilingual parallel sentences.

Furthermore,  the authors demonstrate the usage of MultilingualHolisticBias by analyzing and quantifying MT biases in different translation directions: EN-to-XX, XX-to-EN, and XX-to-XX. For their MT model, they use the open-source NLLB model. The authors found that: 1) EN-to-XX translations are almost 8 BLEU points better when evaluating with masculine references compared with feminine translations; 2) XX-to-EN translations are 4 BLEU better when the source is in the masculine form; 3) XX-to-XX translations vary based on which language was used for source and target. But overall, the translation quality is better when translating from a masculine source to a masculine target.

Moreover, when embedding sentences into a joint multilingual space, they found that masculine targets are closer to the English ambiguous (non-gender) source sentences.





**Questions For The Authors:**

See above.

**Reasons To Accept:**

The paper introduces a valuable multilingual resource that can be used to study and analyze bias in multilingual MT models. The studies that are introduced in the paper demonstrate well how the dataset could be used and its viability. Most of the paper was written clearly and was easy to follow.


**Reasons To Reject:**

I found some parts of the paper to be hard to follow:
* In Figure 4, the authors present results on XX-to-EN translations. I am confused about the results that refer to gendered references in English. The three English templates in the corpus contain descriptors that refer to: 1) third person (as in the first and third templates); 2) first person (as in the second template). Unless the descriptors are gendered, then it will not be possible to tell if the English reference includes a masculine or a feminine reference. So what does the legend of the table stand for here when describing masculine vs feminine English references?

* The figures in the paper were not ordered and do not align with the overall story. I would urge the authors to fix the order of the figures to make the paper easier to read.

**Reproducibility:**

4: Could mostly reproduce the results, but there may be some variation because of sample variance or minor variations in their interpretation of the protocol or method.

**Reviewer Confidence:**

4: Quite sure. I tried to check the important points carefully. It's unlikely, though conceivable, that I missed something that should affect my ratings.

---

> ### Author Rebuttal · Authors · 2023-08-28
>
> The reviewer is totally right, figure 4 should only report feminine source and masculine source. There is no need to report “both” references since the English is neutral. As you can observe from the figure, masculine with one or two references is exactly the same. The same happens with feminine. We are changing the figure accordingly and clarifying in the text.
>
> We have worked on reordering the figures in the paper, specially, moving earlier in the sections the examples. Also, we have added the missing references now.

---

### Official Review · Reviewer_Uryd · 2023-08-04

**Soundness:** 3

**Excitement:**

4: Strong: This paper deepens the understanding of some phenomenon or lowers the barriers to an existing research direction.

**Missing References:**

Beatrice Savoldi, Marco Gaido, Luisa Bentivogli, Matteo Negri, Marco Turchi; Gender Bias in Machine Translation. Transactions of the Association for Computational Linguistics 2021;
and the studies referenced in this survey study.

**Paper Topic And Main Contributions:**

The paper presents Multilingual Holistic Bias, a multilingual extension of the English HolisticBias dataset covering 50 languages. This dataset is then used to particularly quantify the gender bias in machine translation in this study. The key finding is that MT models tend to favour masculine forms across languages (I.e. MT models tend to generate more masculine translations and masculine source sentences tend to be translated with better quality compared with feminine counterparts). This gender bias is also confirmed by examining the multilingual embedding space where masculine translations are usually significantly closer to the English source sentence than feminine translations. The key contributions for the paper are a new and extensive multilingual dataset with a specific focus on gender and demographic biases, and interesting analysis to show that machine translations are biased towards the masculine translations.


**Reasons To Accept:**

- a large-scale multilingual dataset that is manually translated. This will be a valuable resources for evaluating multilingual models on biases.
- The paper proposes a practical framework to quantify the gender biases in MT using the dataset and provides interesting analysis and insights.

**Reasons To Reject:**

- The storyline could be a bit more precise regarding the gender biases across languages. Given that the dataset focuses specifically on toxic descriptors, I think the lessons here from the experiments seem to be that there is a strong association with masculinity and toxicity in many languages rather than in general these language has masculinity biases.

- gender bias in machine translation is not something new. The author(s) might want to better position the work with regard to other studies that investigate biases in machine translation such as "Gender Bias in Machine Translation" Savoldi · 2021


**Reproducibility:**

4: Could mostly reproduce the results, but there may be some variation because of sample variance or minor variations in their interpretation of the protocol or method.

**Reviewer Confidence:**

4: Quite sure. I tried to check the important points carefully. It's unlikely, though conceivable, that I missed something that should affect my ratings.

**Typos Grammar Style And Presentation Improvements:**

Line 203: Among the other languages where have -> Among the other languages where there are

---

> ### Author Rebuttal · Authors · 2023-08-28
>
> Thanks for your comments!
>
> About the storyline, nice point! We now corrected some sentences that did not specify “translation bias”. We are not analising language bias but translation biases for a large scale of translation directions. We clarified when it was not clear.
> About the missing citations, thanks for pointing out the survey of Savoldi. We do a small review of state-of-the-art in the area in the introduction. Since there is a huge amount of literature in studying gender bias in MT, we focus mainly on the works that develop datasets for this topic, which is closer to our work. Given the limitations in space, we are not covering exhaustively even this part. However, it makes sense to cite the survey  you mention.

---

### Meta-Review · Area_Chair_Dbcf · 2023-09-18

**Recommendation:** 5

**Metareview:**

This paper introduces a multilingual extension of the English HolisticBias dataset covering 50 languages, created by manual translation. This dataset is then used to quantify the gender bias in machine translation, and the authors find that MT models tend to favour masculine forms across languages.

The reviewers find that the proposed resource will be valuable for studying and analyzing demographic biases in multilingual machine translation systems and multilingual sentence encoders. Most of the paper was found to be clearly written and easy to follow. Reviewers remarked, however, that some crucial references on biases in MT were missing, and that some parts of the paper could benefit from additional clarifications. The requested changes should be easy to accommodate in the camera-ready version.

---

### Decision · Program_Chairs · 2023-10-07

**Decision:**

Accept-Main

**Comment:**

This paper introduces a multilingual extension of the English HolisticBias dataset covering 50 languages, created by manual translation. This dataset is then used to quantify the gender bias in machine translation, and the authors find that MT models tend to favour masculine forms across languages.

The reviewers find that the proposed resource will be valuable for studying and analyzing demographic biases in multilingual machine translation systems and multilingual sentence encoders. Most of the paper was found to be clearly written and easy to follow. Reviewers remarked, however, that some crucial references on biases in MT were missing, and that some parts of the paper could benefit from additional clarifications. The requested changes should be easy to accommodate in the camera-ready version.